# Role of G-Proteins and GPCRs in Cardiovascular Pathologies

**DOI:** 10.3390/bioengineering10010076

**Published:** 2023-01-06

**Authors:** Geetika Kaur, Shailendra Kumar Verma, Deepak Singh, Nikhlesh K. Singh

**Affiliations:** 1Integrative Biosciences Center, Wayne State University, Detroit, MI 48202, USA; 2Department of Ophthalmology, Visual and Anatomical Sciences, School of Medicine, Wayne State University, Detroit, MI 48202, USA; 3Lloyd Institute of Engineering and Technology, Greater Noida 201306, India

**Keywords:** G-proteins, G-protein-coupled receptors, signaling, cardiovascular diseases, heart failure, atherosclerosis

## Abstract

Cell signaling is a fundamental process that enables cells to survive under various ecological and environmental contexts and imparts tolerance towards stressful conditions. The basic machinery for cell signaling includes a receptor molecule that senses and receives the signal. The primary form of the signal might be a hormone, light, an antigen, an odorant, a neurotransmitter, etc. Similarly, heterotrimeric G-proteins principally provide communication from the plasma membrane G-protein-coupled receptors (GPCRs) to the inner compartments of the cells to control various biochemical activities. G-protein-coupled signaling regulates different physiological functions in the targeted cell types. This review article discusses G-proteins’ signaling and regulation functions and their physiological relevance. In addition, we also elaborate on the role of G-proteins in several cardiovascular diseases, such as myocardial ischemia, hypertension, atherosclerosis, restenosis, stroke, and peripheral artery disease.

## 1. Introduction

Cardiovascular diseases (CVDs) are the foremost cause of morbidity and mortality among all diseases around the world. CVDs are a set of disorders of the heart or blood vessels, or both, consisting of coronary heart disease (CHD), cerebrovascular disease, peripheral arterial disease, myocardial infarction, congenital heart disease, and other cardiac conditions [1,2,3]. CVDs are the primary cause of the socioeconomic burden in the healthcare sector. CVDs were the fastest-growing chronic illness between 2005 and 2015, with a growth rate of 9.2% annually. The World Health Organization (WHO) reports that 31% of all global deaths, i.e., approximately 17.9 million, occur due to CVDs every year [4,5]. Thus, CVDs are significant contributors to reduced life quality and life span in humans [6,7].

The complex pathophysiology of CVDs involves the induction and regulation of various cellular processes governed by intrinsic and extrinsic factors, including mechanical stress, cytokines, and growth factors. These factors are sensed by a wide array of receptors, such as toll-like receptors (TLRs) and nucleotide-binding oligomerization domain-like receptors (NLRs) [8]. The ligands (various regulatory molecules) bind to the receptors on the cell membrane and activate the transmission process of the stimulatory signal inside the cell via a cascade of kinases/phosphatases and second messengers [9]. The interaction of ligands with their receptors, and the subsequent signaling processes, are highly selective due to the specific structure of the receptors and their ligands (key and lock mechanism). G-protein-coupled receptors (GPCRs) are a fundamental signal transduction component associated with cardiovascular remodeling and disease progression [10]. This review discusses the role of G-proteins, their receptors, and the transduction systems in cardiovascular pathologies.

## 2. G-Proteins

G-proteins are a family of specialized proteins that can bind to nucleotides, i.e., guanosine triphosphate (GTP) and guanosine diphosphate (GDP); thus, they are also known as guanine nucleotide-binding proteins [11]. G-proteins are either composed of a single subunit (monomeric) or multiple subunits (heterotrimeric). For example, Ras proteins are monomeric, and G-proteins associated with G-protein-coupled receptors (GPCRs) are heterotrimeric. The monomeric structures, known as small G-proteins/GTPases, consist of an alpha subunit only (Gα), while the heterotrimeric subunit has three different subunits: an alpha (α), a beta (β), and a gamma (γ) subunit. The β and γ subunits form a stable dimeric complex called the βγ complex (Gβγ). The βγ complex is considered a single functional unit and is attached to the plasma membrane by lipid anchors. Humans have 16 Gα, 5 Gβ, and 13 Gγ subunits. Different combinations of these subunits result in a wide range of heterotrimeric G-proteins [12].

Structurally, G-proteins are mainly recognized by their Gα monomers. Based on their sequences and similar activities, these Gα proteins have been classified into four groups, Gαs (stimulatory), Gαi (inhibitory), Gαq, and Gα12/13 [13]. Gαs stimulates adenylyl cyclase to produce cyclic adenosine monophosphate (cAMP), thus activating protein kinase A (PKA) for the regulation of cellular responses [14]. Furthermore, the Gαs family has been divided into two subfamilies, i.e., Gαs (stimulatory, mainly expressed in all types of cells) and Gαolf (olfactory, expressed by sensory neurons). In contrast, Gαi inhibits adenylyl cyclase and dampens intracellular cAMP. The Gαi family has been further divided into seven groups, i.e., Gαi1 Gαi2, and Gαi3 (inhibition, expressed in most types of cells); Gαo (expressed in neurons and transcribed in two spliced variants, GαoA and GαoB); Gαt (transducin), transcribed in two isoforms, Gαt1(expressed in rod cells of the eye) and Gαt2 (expressed in cone cells of the eye); Gαg (gustducin, expressed in the taste receptor cells); and Gαz (expressed in neuronal tissues and platelets). The Gαq family has been further divided into four members, i.e., Gαq (expressed in most types of cells), Gα11(expressed in most types of cells), Gα14 (expressed in lung, liver, and kidney), and Gα15 (expressed in hematopoietic cells). The Gα12 family has been divided into two groups, i.e., Gα12 and Gα13, expressed in most types of cells. As described above, apart from these Gα subunits, the heterotrimeric G-proteins also contain Gβγ protein subunits. Both human and mouse genomes harbor 5 Gβ (Gβ1, Gβ2, Gβ3, Gβ4, and Gβ5) and 12 Gγ genes. Gγ protein subunits are more varied, and their sequence homologies range from 20% to 80%. Gγ protein subunits are widely distributed and expressed [15,16].

Functionally, the family of G-proteins act as molecular switches in the cells, which transmits signals in response to stimuli. The protein becomes active or inactive depending on the binding of the G-protein alpha subunit to either GTP or GDP. In the absence of a signal, GDP attaches to the alpha subunit and forms a G-protein–GDP complex. This arrangement further binds to the GPCR and leads to protein inactivation. Conversely, a signaling molecule changes the conformation of the GPCR and activates the G-protein. GDP is physically replaced by GTP in the activated protein. On GTP’s hydrolysis to GDP, the protein becomes inactive again. G-proteins function by switching on or off via signal–receptor interactions on the cell’s surface [17]. G-proteins possess intrinsic GTPase activity and play dynamic roles in cellular processes such as cell growth, protein synthesis, and membrane vesicle transport [18].

GPCRs are membrane-bound proteins or cell surface receptors in action with various stimuli that include neurotransmitters, hormones, proteins, peptides, small-molecule odorants, pheromones, light, extracellular calcium, protease activity, etc. [19,20]. Generally, GPCRs are odorant/pheromone receptors; however, each mammalian species expresses at least 400 non-odorant GPCRs. GPCR-biased agonism has significantly changed our understanding of GPCR biology in recent years. Biased agonism refers to a condition when a ligand induces a distinctive receptor conformation, culminating in differential coupling to the signal transduction cascade and a different response. These biased GPCR ligands also have important clinical implications, as selectively activating or inhibiting specific signaling cascades could result in more targeted drugs with fewer side effects [21]. Structural studies have shown that GPCRs comprise a transmembrane protein arrangement of seven helices, usually called a heptahelical domain (7TM), and functional extracellular and intracellular loops [22]. This receptor superfamily displays 7TM helical domains linked by alternating three intracellular and three extracellular loops, such as ICL1, ECL1, ICL2, ECL2, ICL3, and ECL3. Five major families in the GPCR superfamily, namely the rhodopsin family, secretin family, glutamate family, adhesion family, and frizzled/taste2 family, have been identified [23]. The heptahelical domain is the only structural similarity or common feature of GPCRs. Receptor activation requires the movement of the transmembrane helices, leading to cavity development on the cytoplasmic side of the receptor. GPCRs have been named based on their ability to bind to G-proteins, which is described as the collision coupling model. The GPCRs bind to Gα subunits in the absence of a ligand. This phenomenon is known as receptor pre-coupling [24]. To date, extensive experimental work has been performed to unravel the molecular aspects of G-proteins’ and GPCRs’ functions such that GPCRs serve as targets for imported drugs to hinder the progression of diseases.

## 3. Regulation of G-Proteins

### 3.1. General Mechanism

In eukaryotes, G-proteins perform a crucial role in controlling multiple signaling pathways. During the inactive phase, the GDP-bound Gα protein subunit is linked strongly to the Gβγ heterodimer [11]. When an agonist binds, signal perception causes a conformational change in the GPCR, resulting in the activation of the GPCR. Upon activation, the GPCR performs as a guanine nucleotide exchange factor (GEF) and increases the exchange of GDP to GTP on the Gα subunit, which further leads to the dissociation of GTP-bound Gα from the Gβγ dimer [25]. Both functional subunits, the GTP-bound Gα monomer and the Gβγ dimer, interact with various effectors to transduce several different cellular signaling pathways. Inactivation arises through the inherent GTPase activity that hydrolyzes the bound GTP and reproduces the GDP-Gα state. GTPase-activating proteins (GAPs), e.g., RGS proteins, bind with the Gα monomer to accelerate GTPase activity. Furthermore, the GDP-bound Gα monomer associates with the Gβγ dimer, thus re-establishing the heterotrimeric complex. Due to the cyclic nature of G-protein signaling, both the activation and inactivation steps have to be synchronized for effective and regular signaling [16]. The entire process represents a G-protein cycle. The mechanism of G-protein signaling is shown in Figure 1.

#### 3.1.1. G-Protein Post-Translational Modification

The post-translational modifications (PTMs) of G-proteins have received relatively little attention, but recent studies have outlined the importance of post-translational modifications of G-protein in the pathogenesis of CVDs. Phosphorylation, ubiquitination, S-nitrosylation, and palmitoylation are the most prevalent in CVDs.

##### G-Protein Phosphorylation

G-protein phosphorylation occurs through kinase-catalyzed transfer and is primarily mediated by receptor tyrosine kinases (RTKs), PKA, and PKC. The serine phosphorylation of Gαq impairs GTP-binding/Gα activation, whereas tyrosine phosphorylation of Gαq promotes Gαq and PLCβ3 activation. Similarly, the phosphorylation of various G-proteins has a widespread application on various signaling pathways, which was discussed in detail by Chakravorty and Assman [26].

##### G-Protein Ubiquitination

Ubiquitin degrades the target proteins via the 26S proteasome and the lysosome, sequentially acting through three distinct enzymes, E1, E2, and E3. The Gα protein ubiquitination regulates Gα subunit trafficking within the cell. The polyubiquitination of Gα leads to its degradation by the proteasomal pathway, whereas its monoubiquitination helps in its trafficking to lysosome [27]. Gαs protein ubiquitination controls epidermal growth factor receptor (EGFR) endosomal sorting [28].

##### G-Protein S-Nitrosylation

S-nitrosylation is a covalent post-translational modification of a protein cysteine thiol by a nitric oxide group. S-nitrosylation occurs in Gαi2 at Cysteine 66, alleviating diabetes-accelerated atherosclerosis [29].

##### G-Protein Palmitoylation

G-protein palmitoylation occurs through palmitate attachment to cysteine residues covalently. Palmitoylation of G-proteins occurs at the alpha and beta subunit, regulating signal transduction [12].

#### 3.1.2. GPCR Regulators

The critical physiological activity of G-proteins is to transduce the signals from GPCRs that work as GEFs for G-proteins. Outer or inner ligand association stimulates GPCRs to enter an active conformational phase that evokes the intracellular linking of G-protein/arrestin proteins [30]. The GPCR-targeted ligands have been categorized into agonists, inverse agonists, and antagonists. The association of agonists with GPCRs leads to an active conformation, thus enhancing the signaling effect. Conversely, inverse agonists impede basal signaling activity via stabilizing inactive GPCR conformations. The neutral antagonists are ineffective towards GPCR conformations; however, they prevent the association of both agonists and inverse agonists [12]. Substantial evidence in the literature suggests that G-proteins bind to ICL2 and ICL3 [13]. An agonist in the extracellular region stimulates a structural rearrangement in the transmembrane’s core portion, which further causes a conformational change in the cytoplasmic intracellular region.

In addition, GPCRs exist in the form of dimers or oligomers. These dimerization and oligomerization forms are involved in the modulation of numerous GPCR activities, such as cell surface targeting, cellular activation, G-protein coupling, and internalization [31,32]. For instance, the dimerization of two protomers for the class-C family of GPCRs is crucial for its biological activity and G-protein stimulation [16].

#### 3.1.3. Non-GPCR Regulators

In contrast to GPCRs, other non-GPCR proteins regulate the heterotrimeric G-proteins, e.g., Ric-8 protein, GPR domains, GBA motif-containing proteins, and RGS proteins [33], which are explained below.

##### Ric-8 Proteins

Mechanistically, the resistance to inhibitors of cholinesterase 8 (Ric-8) proteins interacts with GDP-bound Gα and causes the release of GDP; it leads to the formation of a nucleotide-free Gα intermediate complex and a stable Ric-8. When Gα binds to GTP, it dissociates from Ric-8, which is how the GDP–GTP exchange cycle on Gα is completed [34,35]. The genomic DNA of invertebrates encodes a single ancestral Ric-8 isoform, while the genomes of vertebrates encode two isoforms: Ric-8A and Ric-8B. Each vertebrate isoform controls a particular Gα subset, i.e., Ric-8A regulates Gαi/t, Gαq, and Gα12/13, and Ric-8B regulates Gαs. Ric-8 proteins modulate G-protein signaling via the GEF activity of Ric-8A and Ric-8B and ubiquitous chaperone activity [35,36]. According to estimates, the heritability of the resting heart rate is between 20% and 60% [37,38]. Small variations in RIC-8B expression affect basal cAMP signaling, affecting the resting heart rate. Recently, genome-wide association studies have reported a relationship between a single-nucleotide polymorphism in a region of the genome encoding the RIC-8B gene and heart rate [39]. The RFX4 gene’s rs2067615 variant on chromosome 12 increased the heart rate by 0.3 beats per minute per allele. A few genes are located within 500 kilobases of this locus including RFX4, RIC-8B, C12orf23, MTERFD3, CRY1, POLR3B, and TCP11L2. RIC-8B, CRY-1, POLR3B, and TCP11L2 are expressed in cardiac cells [40].

##### GPR Domains

The proteins containing G-protein regulator (GPR) domains are the second most significant class of non-GPCR regulators. GPR domains comprise 25-amino-acid-long stretches, also known as the GoLoco domain. GPR domains bind to the inactive state of GDP-bound Gαi/o. These GPR proteins restrict the nucleotide exchange in Gα and compete with Gβγ subunits for Gα binding. When Gα is bound to a GPR domain instead of Gβγ subunits, Ric-8A stimulates Gα by promoting the exchange of GDP for GTP. In addition, GPR-domain-containing proteins are harbored by some activators of G-protein signaling (AGS) proteins. For example, AGS3 has four GPR domains associated with the Gαi family [41].

##### GBA Motif

A structurally defined component connected to the activity of GEF has been elucidated as the Gα binding and activating (GBA) motif [42,43]. The GBA motif contains 30 residues in its sequence. It includes Gα-interacting-vesicle-associated protein (GIV or Girdin), DAPLE, NUCB1, CALNUC, and NUCB2, which interact and stimulate Gαi member proteins [42]. GIV and DAPLE are well-characterized proteins, and despite their moderate GEF activity, many in vitro experiments have confirmed that the GBA motif accelerates the nucleotide exchange rate, which further activates the Gα subunit [44,45]. Moreover, GIV could not be associated with Gα in the active Gα-GTP-bound form. This characteristic of GEFs confirms that GIV only binds with the inactive Gα-GDP-bound form and consequently stimulates Gα signaling. In a study, Garcia-Marcos et al. (2009) reported that GIV dissociates Gβγ from the Gαi-βγ heterotrimer complex and accelerates Gβγ-dependent signaling in cells [46]. However, it is yet to be confirmed whether GIV can directly stimulate a Gαi-βγ heterotrimer. Overall, the GBA-motif-containing proteins impact cell behavior. Their abnormal expression is associated with different diseases, such as tumors, liver fibrosis, cancer, diabetes, and pathologic neovascularization [47]. The disruption of GIV and DAPLE association with G-proteins is evolving as a promising therapeutic approach [48].

##### RGS Proteins

Regulators of G-protein signaling (RGS) proteins increase the GTPase activity and deactivate the signal started by GPCRs. A large family of RGS proteins consists of 37 members with a conserved RGS homology domain. RGS proteins play vital roles in many physiological and pathological conditions [49]. Mechanistically, RGS proteins bind with active Gα subunits in GTP-bound form to enhance GTP hydrolysis. RGS proteins accelerate the GTP hydrolysis rate by up to 2000 times, changing the Gα–GTP complex conformation and improving the GTPase efficiency of Gα. The transition state in GTP hydrolysis can be imitated by the Gα–GDP–A1F4 complex. The GTPase-activating protein (GAP) activity of RGS proteins may be involved in altering the affinity of RGS for Gα or stabilizing the catalytic conformation of Gα. While the typical mode of action of RGSs is to associate with Gα subunits, some of them can also bind directly with effectors, such as the RGS2 protein, which can inhibit type V adenylyl cyclase [50]. RGS proteins also regulate various critical cardiac processes, and abnormal changes in their expression are frequently associated with cardiovascular system dysfunction. For example, RGS2 deficiency increases angiotensin II (AngII) type 1 (AT1) receptor signaling, leading to hypertension [51], and RGS2 and RGS14 regulate cardiac remodeling [52,53]. Among other disorders, RGS proteins are also involved in heart failure and drug-induced cardiac damage [54,55]. Cardiovascular disease has been associated with changes in RGS expression levels, suggesting that aberrant RGS protein expression may contribute to pathophysiology. RGS proteins and GPCRs work in a bi-directional manner, with RGS regulating GPCR activity and GPCR activation changing the expression of RGS proteins. For instance, the AT1 receptor controls the expression of RGS2 [56], RGS10 [57], and RGS14 [53], all of which control the effects of the AT1 receptor. Similar to this, isoproterenol, an agonist of the 1 and 2 adrenoceptors, increases the expression of RGS5 [58], and the activation of the lysophospholipid sphingosine 1-phosphate (S1P) receptor controls the expression of RGS2 and RGS16 in vascular smooth muscle cells [59].

In the recent past, many genetic models have been developed, signifying the clinical importance of G-proteins and GPCRs in different diseases, including diabetes, cardiovascular disease, various types of cancer, and disorders of the central nervous system, which may open up new research avenues for the development of novel drugs/inhibitors.

## 4. Role of G-Proteins in Cardiovascular Diseases

G-proteins and their receptors are extensively expressed in the cardiovascular system and are involved in the pathophysiology of cardiovascular diseases, as summarized in Table 1. Much of the G-protein signaling is mediated by several signaling effectors, such as adenylyl cyclase (AC), Ras homology (Rho), cell division cycle 42 (cdc42), phospholipase C (PLC), and SRC, which contribute to various diverse cellular processes (Figure 2).

### 4.1. Heart Failure

Heart failure (HF) is a medical condition that develops when the heart cannot pump enough blood to meet the body’s needs. HF affects over 6.5 million American adults and results in an annual healthcare burden of USD 30 billion [2]. Heart failure (HF) is a common and serious condition with substantial morbidity and mortality rates. HF is associated with structural or functional abnormalities caused by cardiomyocyte enlargement and hypertrophic growth. Several intrinsic and extrinsic stimuli, such as stress, cytokines, and growth factors, are sensed by cardiomyocyte receptors such as GPCRs, causing detrimental effects [17]. Inotropic and chronotropic hyporesponsiveness to adrenergic stimulation, as well as a decrease in Gs-alpha proteins, or an increase in Gi-alpha proteins, lead to congestive heart failure [67]. In these conditions, there is strong sympathetic activation, which causes a decrease in beta-adrenergic activity. In terms of membrane receptors, beta1 receptors are downregulated, and beta2 receptors are uncoupled. Even without beta-adrenergic receptor downregulation, an increase in Gi proteins can suppress adenylate cyclase activity. These results demonstrate that the Gi protein desensitization of adenylate cyclase can serve as an essential pathophysiological mechanism in the development of compensated cardiac hypertrophy to HF, because cardiac hypertrophy is a major predictor of HF [68]. Additionally, similar changes can be observed with aging [69].

Acetylcholine-dependent activation of cardiac potassium channels regulates heart rate. G-protein beta signaling mediates the activated muscarinic receptor-induced stimulation of cardiac potassium channels. The G-protein-coupled inwardly rectifying potassium channels (GIRKs) encode cardiac potassium channels. GIRK1 and GIRK4 are two members of the GIRK family located in the heart [70,71]. The cell membrane becomes hyperpolarized due to an interaction between the activated G-protein subunits (G) released by GPCRs and GIRK channels for potassium ion permeability. The neuron cannot fire action potentials quickly when it is hyperpolarized, which slows the heartbeat [72]. It is also emphasized that SUMOylation, O-GlcNAcylation, acetylation, and phosphorylation all play a role in the pathogenesis of HF and cardiac remodeling [73].

GPCRs are essential in numerous physiological processes and therefore are targets of pharmaceutical therapeutics. For instance, the activation of β-adrenergic receptors (βARs) and Ang II type 1 receptors (AT1Rs) results in myocyte death and adverse cardiac remodeling, as well as an increased heart workload [74]. To transmit signals, AT1Rs couple to Gα_q,_ Gβγ, and β-arrestin and form AT1R–β-arrestin complex. Experimental evidence in the literature suggests that the activation of β-arrestin and blocking of G-proteins downstream of AT1R may provide additional benefits compared to Ang II blockers alone. Therefore, receptor blockers such as β-blockers, Ang II receptor blockers, and ACE inhibitors are widely used in the treatment of HF [17]. AT1R–β-arrestin-biased ligands, including TRV120027 and TRV120023, have demonstrated advantages over Ang II blockers in cardiac and renal function. While TRV120023 inhibits Ang II-induced cardiac hypertrophy and supports cardiomyocyte survival following ischemia injury, TRV120027 stimulates vasodilation by blocking the G-protein pathway and improves cardiac contractility [17,75,76]. As a result, these β-arrestin-biased ligands offer promising new HF therapies.

### 4.2. Myocardial Ischemia

Myocardial ischemia occurs after an imbalance between the oxygen supply and demand in the myocardium. This imbalance is responsible for myocardial infarction, arrhythmias, cardiac dysfunction, and sudden death. The obstruction of coronary blood flow due to thrombosis, coronary stenosis, and the hypercontraction of epicardial and coronary arteries lead to several clinical ischemic manifestations. Generally, GPCRs are essential for normal cellular function; however, sustained signaling may cause damage to the cardiac cells and functioning. The adrenergic receptor (AR) on the cardiomyocyte plays an important role in myocardial ischemia. These receptors are the principal regulators that activate adenylyl cyclase, enhancing cAMP and mediating cellular processes [77]. The adrenergic receptors are of two types: β1 and β2. Under normal conditions, β1AR, the most abundant in cardiomyocytes, comprises 80% of receptors, whereas β2AR comprises approximately 20%. In a diseased state, the stoichiometry changes to 60:40. The underlying mechanisms responsible for the loss of adenylyl cyclase functioning consist of reversible and irreversible phases. In the reversible phase, uncoupling of G-protein receptors and allosteric alteration of the catalytic subunit is observed. Meanwhile, in the irreversible step, free radicals cause alterations in adenylyl cyclase, which last for more than 30 min in ischemia. This functional imbalance of G-proteins is commonly observed in acute myocardial ischemia. β1 adrenergic receptor activate cardiac transduction pathways, leading to early myocyte hypertrophy, cardiac hypertrophy, and interstitial fibrosis when overexpressed, while β2AR signaling has cardioprotective effects. Therefore, beta-blocking agents are effectively used to treat myocardial ischemia [78].

Several modifications cause the ischemic myocardium’s loss of adenylate cyclase function. The reversible phase of this process is characterized by G-protein-receptor decoupling and potential allosteric changes in the catalytic subunit, which elevate the calcium levels in a compartment next to the enzymatic activity. Free radicals are assumed to be substantial, if not entirely, responsible for the permanent modification of adenylate cyclase function seen in ischemia lasting more than 30 min (global normothermic ischemia) [79]. Numerous investigations have demonstrated that rather than decreasing, the density of β adrenergic receptors increase in the plasmatic membranes of ischemic hearts. Acute myocardial ischemia causes the loss of high-energy phosphates, which stops beta-receptor coupling and signal transduction. As a result, there are more beta receptors on the cellular surface because exterior phenomena predominate over internal ones. During the early desensitization phase of the first pathophysiological stage of myocardial ischemia, the uncoupling of G-protein receptors is entirely stopped. Exogenous catecholamine administration to ischemic hearts cannot counteract this effect [80,81].

The activation of G-proteins by β adrenergic receptors increases enzymatic activity, whereas the activation of Gi proteins by M2 muscarinic receptors and A1 adenosine receptors decreases enzymatic activity. Adenylate cyclase inhibition lowers both its basal and stimulated activity. As a result, adenylate cyclase’s responsiveness to stimulating hormones is reduced by a tonic rise in Gi protein inhibitory activity. The adenylate cyclase system, on the other hand, becomes more responsive or sensitive when tonic inhibition is lost. In many acute myocardial ischemia models, G-proteins have been demonstrated to be functionally unbalanced. Gi protein levels rapidly lose functional activity following ischemic damage, but Gs protein levels are stable for a considerable time. Further research is necessary to understand the molecular mechanisms causing this functional impairment. Many experimental models of acute myocardial infarction (AMI) showed a general decrease in adenylate cyclase activity as ischemia progressed. Adenylate cyclase activity is independent of β adrenergic receptors and G-proteins, according to recent investigations on the development of acute ischemia. This activity is linked to protein kinase activation [82]. The succinylation, phosphorylation, SUMOylation, acetylation, and glutathionylation of G-proteins are all involved in the formation and progression of I/R injury and the regulation of cardiac repair [73].

### 4.3. Hypertension

High blood pressure, sometimes known as hypertension, is a complicated multifactorial condition. The force exerted by blood causes pathological changes in the arteries and arterioles, resulting in severe conditions such as target organ damage, atherosclerosis, and kidney diseases [83]. Hypertension is also considered a silent killer [84]. A delicate balance between vasoconstrictors and vasodilators is essential for maintaining blood pressure [85]. GPCRs function in vasodilation and vasoconstriction. Beta-adrenergic impairment, with alterations in receptor–G-protein interaction, is primarily responsible for the development of hypertension [86]. Mechanistically, GPCR ligands such as angiotensin II, endothelin 1, and vasopressin, via Gαq, stimulate the activity of phospholipase C-β to form inositol-1,4,5-trisphosphate (IP3) and diacylglycerol (DAG). The pathogenesis of hypertension is also influenced by the acetylation, phosphorylation, O-GlcNAcylation, SUMOylation, and S-glutathionylation of G-proteins [73].

The binding of IP3 to its respective receptor leads to calcium efflux, while DAG promotes calcium influx by activating PKC [87]. These calcium levels increase myosin light chain kinase (MLCK) activity. MLCK activity is counter-balanced by phosphorylated MLC, thus acting in constriction. In addition, the activation of Rho kinase pathways also regulates blood pressure [88]. Conversely, adrenaline acts as a vasodilator that binds to its corresponding receptor to stimulate adenylyl cyclase via Gαs. The resulting cAMP formation activates PKC. This crosstalk between calcium efflux and influx leads to myosin–actin filament interactions for vascular smooth muscle cell contraction and relaxation. The alteration in ligand–GPCR exchange impairs the vascular smooth muscle cell contraction and relaxation processes, and altered adenylyl cyclase activity is responsible for high blood pressure [89].

G-proteins regulate signal transduction systems such as adenylyl cyclase/cAMP and phospholipase C (PLC)/phosphatidyl inositol turnover (PI), cardiovascular performance, and functions, such as arterial tone and responsiveness. Reports in the literature have shown that inhibitory G-proteins regulate the expression of G-proteins, stating Gi proteins as an essential contributing factor to hypertension. It has been demonstrated that elevated amounts of vasoactive peptides, such as angiotensin II (AngII), contribute to increased Gi protein expression, adenylyl cyclase signaling, and elevated blood pressure. Furthermore, increased oxidative stress in hypertension caused by Ang II may be responsible for the increased expression of Gi proteins observed in hypertension [90].

Alpha-adrenergic receptors are crucial for controlling blood pressure. The α-adrenergic receptor (αAR) family is composed of the α1AR (α1A, α1B, α1D) and α2AR (α2-adrenergic receptor; α2A, α2B, α2C) subfamilies [91]. Catecholamines bind to and activate αARs, similar to βARs. In subfamilies, α1ARs, expressed in the heart and cardiomyocyte, couple to Gαq to activate PLC to generate second messengers to increase intracellular Ca^2+^ levels [92]. The α1ARs also perform cardioprotective functions, such as hypertrophy, increased contractility, and decreased apoptosis [93]. Thus, it is recommended to exercise caution when using α1AR antagonists as drugs for the treatment of hypertension, as doxazosin and prazosin drugs have been associated with an increased incidence of heart failure [94].

### 4.4. Atherosclerosis

Atherosclerosis is a progressive disease involving the hardening and thickening of the mid- and large-sized arteries due to the accumulation of modified lipids in the arterial vessel wall and the formation of atheromatous plaques [95]. The atheromatous plaque consists of modified lipoproteins, foam cells, leukocytes, migrated vascular smooth muscle cells (VSMCs), necrotic cores, and calcified regions [96]. In the disease condition, endothelial dysfunction is the primary step, leading to endothelium impairment. An impairment in endothelium-dependent vasoconstrictors such as endothelin (ET) and thromboxane (Tx) and vasodilators can lead to abrogative coronary vascular tone [97]. ET binds to either of the receptors, ETA or ETB. Among them, ETA receptors have significance in the cardiovascular system. ETA receptors activate Gαq, resulting in the formation of IP3 and activation of MAPK signaling.

Moreover, these ETA receptors may inhibit adenylyl cyclase via Gi coupling. Based on this mechanism, ET receptor antagonists such as bosentan, sitaxentan, macitentan, or ambrisentan have shown a cardioprotective role. Further, G-protein-coupled receptor 124 (GPR124), an orphan receptor, plays a significant role in the development and progression of atherosclerosis by activating nitrosative stress and NLR family pyrin domain containing 3 (NLRP3) inflammasome signaling. In a study, Gong et al. (2018) suggested that GPR124 manipulation in the endothelium might lead to the delayed progression of atherosclerosis in an animal model [98]. This receptor can be used as a potential therapeutic target for atherosclerotic pathologies. In a recent review, Zhou et al. (2019) discussed the role of lysophosphatidic acid (LPA) and its receptors in the pathophysiology of atherosclerosis [99]. LPA is generated during the metabolism of lipids and accelerated by activated platelets, an essential step in atherosclerotic initiation and development, respectively. The extended role of GPCR transactivation of tyrosine and serine/threonine kinase growth factor receptors have been recognized. For instance, LPA-enhanced monocyte chemotactic protein-1 expression is mediated via a Gαi-RhoA-ROCK-NF-κB-dependent signaling pathway [100]. Consequently, the LPA receptor might be a beneficial therapeutic agent to halt the progression of atherosclerosis. GPCR agonists and antagonists are used to treat various cardiovascular conditions, and the currently available drugs used for hypertension, heart failure, and atherosclerosis are listed in Table 2.

Gs and Gi protein changes are linked to coronary artery disease. However, the relationship between Gs and Gi proteins needs to be clarified. Several studies have found that patients with coronary artery disease have either decreased Gs proteins and normal Gi proteins or increased Gi proteins and normal Gs proteins. It is critical to note that higher Gi protein levels are associated with more severe coronary artery deterioration than lower Gs protein levels [108]. Members of the Ras protein superfamily, such as Rho proteins, play a role in the pathophysiology of atherosclerosis. The interaction of cytokines, chemokines, and immune cells such as monocytes, macrophages, lipid droplets, and foam cells causes atherosclerosis. The Rho GTPase regulates and acts as a molecular switch for ROCK interaction and GTP-bound conformation in these atherosclerosis-related cells. On the other hand, GTPase-activating proteins and guanine nucleotide dissociation inhibitors inactivate Rho GTPase [109]. The acetylation, phosphorylation, nitrosylation, SUMOylation, and S- glutathionylation of G-proteins contribute to atherosclerosis [73].

### 4.5. Stroke

Stroke, or cerebral ischemia, is a leading cause of global mortality. It occurs due to ischemic insults and the blockage of a major cerebral artery due to the formation of a thrombus or an embolism. Loss of blood flow and tissue death occurs due to oxygen and glucose deprivation [110,111]. Evidence in the literature suggests a substantial role of GPCRs in the pathophysiology of stroke. More than 90% of GPCRs are expressed in the brain, and their roles have been identified in several processes, including immune regulation, cognition, synaptic transmission, and pain. GPCR ligands, such as oxytocin, serotonin, muscarinic acetylcholine, and cholinergic, play a vital role in activating intracellular signaling pathways [112,113]. For instance, serotonin is a neurotransmitter with both a protective and detrimental role in ischemic brain injury. All the serotonin receptors are coupled to Gαi/o, Gαs, and Gαq/11 proteins [114]. Activated serotonin receptors stimulate Gi/Go proteins, which leads to the inhibition of adenylyl cyclase, thereby reducing cAMP formation. This process reduces the phosphorylation of ion channels and neuronal excitation [115]. Studies have shown the neuroprotective benefits of serotonin agonists such as piclozotan and repinotan against ischemic brain injury [116].

Another example is dopamine, a neurotransmitter in the brain that controls locomotor activity, learning, and memory processes, along with positive reinforcement [117]. Dopaminergic receptors are of five types: D1–D5. D1 and D5 are coupled to Gs proteins, which further activate adenylyl cyclase and PKA, while other receptors are coupled to Gi/o proteins and inhibit adenylyl cyclase and PKA [118,119]. Thus, some GPCR agonists and antagonists have neuroprotective benefits, and their receptors are considered drug targets [120].

### 4.6. Peripheral Artery Disease

Peripheral artery disease (PAD) is the narrowing or blockage of vessels due to fatty plaque build-up, i.e., an atherosclerotic disease that affects blood vessels in the arms and legs and excludes coronary circulation [121]. The role of GPCRs in atherosclerosis is similar to that of GPCRs in PAD. GPCRs, such as adenosine receptors, are expressed in human organs. There are four significant subtypes of adenosine receptors: A1R, A2aR, A2bR, and A3R [122]. A1R and A3R function through Gi, whereas A2R couples to Gs. Stimulation of adenosine receptors releases Gβγ subunits, which play an essential role in cell growth and vascular remodeling. A1R interacts with PLC, influencing IP3 and calcium release. Thus, it is directly or indirectly involved in modulating calcium potassium channels [17,123]. Therefore, adenosine agonists and antagonists may have a cardioprotective role in therapeutics [124].

### 4.7. Restenosis

Restenosis is the re-narrowing of the arterial lumen following a vascular intervention intended to treat lesions, such as direct repair (patch angioplasty, endarterectomy) and intraluminal repair (balloon angioplasty, atherectomy, stent angioplasty). Restenosis also results from thrombosis, elastic recoil, remodeling, and intimal hyperplasia [125,126]. In restenosis, G-protein signaling is transient and followed by desensitization and receptor internalization. Beta-arrestin (βarr) is abundantly expressed in cardiac muscles in two isoforms: βarr1 and βarr2 (arrestin-2 and -3, respectively) [127].

β-arr binds to the receptor’s phosphorylated residues and at the intracellular core of the heterotrimeric G-protein binding site. This results in the steric blocking of G-protein binding to the receptor. Thus, β-arr recruitment leads to the uncoupling of G-proteins and signaling desensitization. In addition, β-arr recruits clathrin-coated pit (CCP) proteins such as clathrin heavy chain and the clathrin adapter protein-2 (AP2), which is followed by desensitization and receptor internalization [128,129]. Besides this, GPCR agonists such as angiotensin II and alpha-thrombin have also been implicated in restenosis [130].

In restenosis, heterotrimeric G-proteins such as Gβγ are involved in the activation of mitogen-activated protein (MAP) kinases and proliferation of vascular smooth muscle (VSM) cells. In addition, βarrestin (βarr)-1 and -2 (βarrs) are universal GPCRs expressed abundantly in the myocardium and act as molecular switches for G-protein-dependent to G-protein-independent signaling processes. βARs and AT1R have cardioprotective benefits as these molecules attenuate apoptosis [131].

## 5. Conclusions

G-protein signaling regulates various pathological responses, widely expressed throughout the body. Thus, they are considered significant in the physiological regulation of cardiac function. G-proteins activate several signaling pathways that regulate various cellular processes. Due to their role in cellular signal transduction, G-proteins and GPCRs are considered major drug targets for treating cardiovascular disease. Advances in our understanding of G-proteins’ structure, functions, and signaling mechanisms have contributed to developing a drug with enhanced specificity and efficacy. However, the reversible and irreversible phases and the stimulation of Gs and Gi proteins with different receptor sub-types, both GPCR agonists and antagonists, have therapeutic potential, which makes this a controversial subject. Therefore, new GPCR ligands and mechanisms of action need to be studied and discovered to obtain new therapeutic strategies and medicines.

## Figures and Tables

**Figure 1 bioengineering-10-00076-f001:**
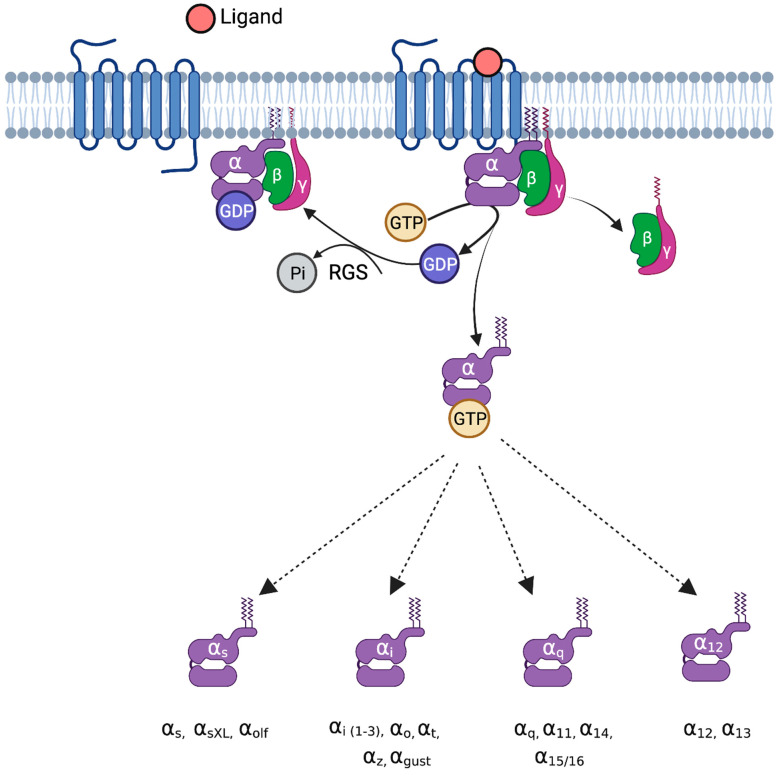
G-proteins and their mechanism of activation. Upon ligand binding to G-protein-coupled receptors, the exchange of GDP for GTP takes place on the Gα subunit, leading to decreased binding affinity for Gβγ. The GTP-bound Gα interacts with several downstream effectors and leads to various pathophysiological functions. Based on sequence homology and activity, Gα proteins are classified into four groups: G-alpha-s (stimulatory), Gαi (inhibitory), Gαq, and Gα12/13. Gαs is divided into Gαs, GαXL (extra-long), and Gαolf (olfactory). The G-alpha-i family is divided into Gαi (1-3), Gαo, Gαt, Gαz, and Gαgust. The Gαq family is divided into Gαq, Gα11, Gα14, and Gα15/16. The Gα12 family is divided into Gα12 and Gα13. RGS, regulator of G-protein signaling.

**Figure 2 bioengineering-10-00076-f002:**
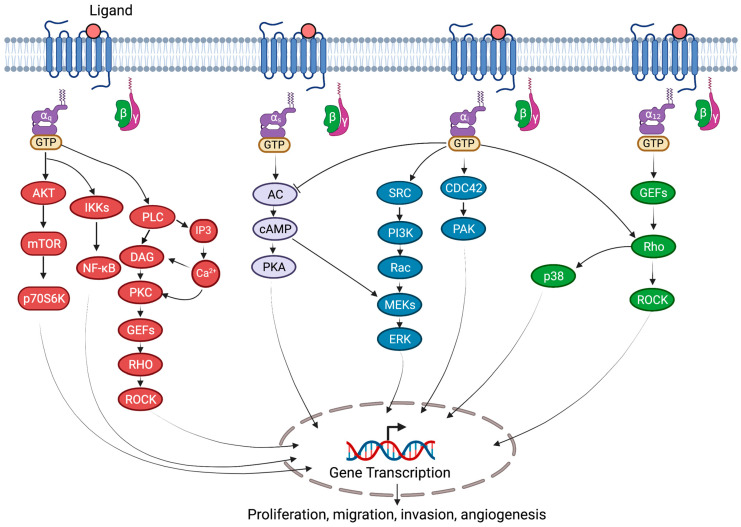
G-protein-coupled signaling in cardiovascular diseases. Various G-proteins, Gαs, Gαi, Gαq, and Gα12/13, activate several downstream diverse signal transduction pathways. Gαs proteins stimulate AC-cAMP-mediated activation of ERK and PKA signaling. Gαq activates phospholipase C (PLC), which cleaves phosphatidylinositol-4,5-bisphosphate (PIP2) into diacyl glycerol (DAG) and inositol 1,4,5-trisphosphate (IP3), which leads to calcium mobilization and protein kinase C (PKC) activation. Furthermore, PKC activation induces Rho guanine nucleotide exchange factors (RhoGEFs) and mitogen-activated protein kinase (MAPK) signaling. Gαq also activates AKT-mTOR and NF-κB signaling pathways. Gαi activates SRC-PI3K-RAC-MEK-ERK and CDC42-PAK signaling. Gα12/13 activate RhoGEFs-RHO-dependent p38 MAPK and ROCK signaling. AC, adenylyl cyclase; CDC42, cell division cycle 42; cAMP, cyclic AMP; ERK, extracellular signal-regulated kinase; IKKs, IκB kinases; mTOR, mammalian target of rapamycin; p70S6K, ribosomal protein S6 kinase; NF-κB, nuclear factor-κB; PKA, protein kinase A; PAK, p21/CDC42/RAC1-activated kinase; ROCK, Rho-associated coiled-coil containing protein kinase; Rho, RAS homology.

**Table 1 bioengineering-10-00076-t001:** Different GPCRs, ligands, and their locations in the body.

GPCR	Ligand	Location	Reference
*Adrenergic receptor*	Norepinephrine	Cardiomyocytes	[60]
*Angiotensin II receptor*	Angiotensin II	Endothelial cells	[61]
*Endothelin receptor*	Endothelin I	Blood vessel	[62]
*Adenosine receptor*	Adenosine	Heart and brain	[63]
*LPA receptor*	Lysophosphatidic acid	Heart and brain	[64]
*Serotonin receptor*	Serotonin	Cardiac cells	[65]
*Muscarinic receptors*	Acetylcholine	Cardiac myocytes	[66]

**Table 2 bioengineering-10-00076-t002:** GPCR-targeted drugs used for cardiovascular diseases.

Drug(s)	Condition	Target	Reference
Irbesartan (Avapro)	Hypertension	Angiotensin II	[101]
Losartan	Hypertension	Angiotensin II	[102]
Vasomera (PB1046), vasoactive intestinal peptide	Hypertension	VIP and PACAP receptor family: VIPR1, VIPR2	[103]
Serelaxin	Heart failure	Relaxin receptor: RXFP1, RXFP2	[103]
TRV120027	Heart failure	Angiotensin II	[104]
Plozalizumab	Atherosclerosis	CCR2	[105]
Alfuzosin, Terazosin	Hypertension	Adrenoreceptor: Alpha-1	[106]
Clonidine, Bisoprolol, Betaxolol	Hypertension	Adrenoreceptor: Alpha-2	[106]
Metoprolol, Atenolol	Hypertension	Adrenoreceptor: Beta-1	[106]
Atropine, Isoproterenol Forskolin Acetylcholine	Heart rate reduction	Muscarinic receptors: Gαq (M1, M3, M5), Gαi (M2, M4)	[17,107]

## Data Availability

Data available in a publicly accessible repository.

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
