# Peer review of "Role of G-Proteins and GPCRs in Cardiovascular Pathologies"

_bioengineering, 2023, doi:10.3390/bioengineering10010076_

Round 1

Reviewer 1 Report

This review aims to consider the role of G proteins in relation to cardiovascular disease. I assume it is aimed at non-specialists. It does have some relevant material but I am surprised that some areas are not covered and equally I struggle to see the relevance of some material that is included. At times the authors' grasp of pharmacology seems somewhat strained and there are places where the English is very strange.

Major points

1) I am very surprised that heart failure is not included. This is a major cause of death involving the cardiovascular system and the first stage treatments are either beta blockers or drugs that target the angiotensin system, including angiotensin receptor blockers. This seems to me to be a major omission. 

2) I am somewhat surprised that there are no mention of alpha adrenergic receptors in the review; it is potentially misleading to consider that all the effects of the adrenergic system are mediated by beta receptors. Alpha 1 adrenoreceptors are major drivers of peripheral vasoconstriction; central alpha 2 receptors are important in overall control of blood pressure.

3) Section 3.1.2, "Non-GPCR regulators" lacks any linkage to the cardiovascular system. Lines 197-8 state "Their abnormal expression is associated with different diseases such as tumors, liver fibrosis, cancer, diabetes, and pathologic neovascularization". Only the last of these conditions has any possible linkage to the topic of this review. Can the authors show examples of where the proteins they discuss in this section play a role in vascular or cardiac tissues? 

4) I cannot see any mention of regulation of ion channels by G proteins. Particularly for the heart, this is very important but K+ channels are also important for the contractile state of myocytes in blood vessels.

5) Whilst I appreciate the focus of this review is on G proteins, I would have though the authors should at least acknowledge the potential of biased signalling at GPCRs, so that either the G protein or beta arrestin pathways are stimulated. The work of Lefkowitz and others has shown that the clinically important beta blocker, carvedilol is additionally an agonist at the beta arrestin pathway and that this may be therapeutically important.

6) Section 4, whilst nominally about G proteins, has far more to say about G protein coupled receptors. Possibly the authors need to change the title of their manuscript to include these as well as G proteins. What is known about how the expression of the different G proteins changes in the conditions that the authors discuss?

Minor points

1) Line 4; is the list of authors complete or should there be another name after "and 2"?

2) Lines 87, 107. The authors describe how inactive G proteins can associate with GPCRs. This is possible, but collision coupling between inactive G proteins and GPCRs is, I suspect, more common; Calebiro D, Koszegi Z, Lanoiselée Y, Miljus T, O'Brien S.  Physiol Rev. 2021 Jul 1;101(3):857-906.

3)  Lines 147-150. This is muddled. Ligand association drives/stabilises distinct conformations of the GPCR, either active or inactive. It is a neutral antagonist that simply prevents the binding of another agent; the qualifying word is important.

4) Lines 151-2; "G-proteins bind to ICL2 and ICL3, i.e., N and C- terminals of GPCRs". No, these are intracellular loops, they are nothing to do with the N and C-terminals of a GPCR.

5) Line 158 protomer, not promoter

6) Line 176 stretches, not starches (!)

7) Line 229 tris... not tirs...

8) Line 249, Beta adrenergic receptors and they are found on the cell surface, not the nuclear membrane

9) Line 272, peptides, not proteins

10) Line 278, I do not understand the phrase "thus acting in constriction"

11) Lines 322-3 "muscarinic acetylcholine, cholinergic" do not make sense as descriptions of GPCR ligands; acetylcholine alone is a ligand

12) Line 331 dopamin acts a neurotransmitter in the brain, not a catecholamine

13) Lines 335-6 "Thus, few GPCRs agonists and antagonists have neuroprotective benefits and are considered drug targets". I think this should read "Thus, some GPCRs agonists and antagonists have neuroprotective benefits and their receptors are considered drug targets"

14) Line 369 "considered" not "ruminated"

Author Response

# Reviewer 1:

  • I am very surprised that heart failure is not included. This is a major cause of death involving the cardiovascular system and the first stage treatments are either beta blockers or drugs that target the angiotensin system, including angiotensin receptor blockers. This seems to me to be a major omission. 

Answers: In response to Reviewer #1’s suggestions, we have now incorporated a new section on heart failure in the revised manuscript (please refer to Pages 8 & 9, lines 1049-1312).

  • I am somewhat surprised that there are no mention of alpha adrenergic receptors in the review; it is potentially misleading to consider that all the effects of the adrenergic system are mediated by beta receptors. Alpha 1 adrenoreceptors are major drivers of peripheral vasoconstriction; central alpha 2 receptors are important in overall control of blood pressure.

Answers: In response to Reviewer #1’s suggestions, we have incorporated it in the revised manuscript (please refer to Page 11, lines 1889-1898).

  • Section 3.1.2, "Non-GPCR regulators" lacks any linkage to the cardiovascular system. Lines 197-8 state "Their abnormal expression is associated with different diseases such as tumors, liver fibrosis, cancer, diabetes, and pathologic neovascularization". Only the last of these conditions has any possible linkage to the topic of this review. Can the authors show examples of where the proteins they discuss in this section play a role in vascular or cardiac tissues? 

Answers: In response to Reviewer #1’s suggestions, we have incorporated it in the revised manuscript (please refer to Page 5, lines 512-520; Pages 6 & 7, lines 811-972).

  • I cannot see any mention of regulation of ion channels by G proteins. Particularly for the heart, this is very important but K+channels are also important for the contractile state of myocytes in blood vessels.

Answers: In response to Reviewer #1’s suggestions, we have now included it in the revised manuscript (please refer to Pages 8 & 9, lines 1067-1303).

  • Whilst I appreciate the focus of this review is on G proteins, I would have though the authors should at least acknowledge the potential of biased signalling at GPCRs, so that either the G protein or beta arrestin pathways are stimulated. The work of Lefkowitz and others has shown that the clinically important beta blocker, carvedilol is additionally an agonist at the beta arrestin pathway and that this may be therapeutically important.

Answers: In response to Reviewer #1’s suggestions, we have included it in the revised manuscript (please refer to Page 9, lines 1336-1341).

  • Section 4, whilst nominally about G proteins, has far more to say about G protein coupled receptors. Possibly the authors need to change the title of their manuscript to include these as well as G proteins. What is known about how the expression of the different G proteins changes in the conditions that the authors discuss?

Answers: In response to Reviewer #1’s suggestions, we have now modified the title.

Minor points

1) Line 4; is the list of authors complete or should there be another name after "and 2"?

2) Lines 87, 107. The authors describe how inactive G proteins can associate with GPCRs. This is possible, but collision coupling between inactive G proteins and GPCRs is, I suspect, more common; Calebiro D, Koszegi Z, Lanoiselée Y, Miljus T, O'Brien S.  Physiol Rev. 2021 Jul 1;101(3):857-906.

3)  Lines 147-150. This is muddled. Ligand association drives/stabilises distinct conformations of the GPCR, either active or inactive. It is a neutral antagonist that simply prevents the binding of another agent; the qualifying word is important.

4) Lines 151-2; "G-proteins bind to ICL2 and ICL3, i.e., N and C- terminals of GPCRs". No, these are intracellular loops, they are nothing to do with the N and C-terminals of a GPCR.

5) Line 158 protomer, not promoter

6) Line 176 stretches, not starches (!)

7) Line 229 tris... not tirs...

8) Line 249, Beta adrenergic receptors and they are found on the cell surface, not the nuclear membrane

9) Line 272, peptides, not proteins

10) Line 278, I do not understand the phrase "thus acting in constriction"

11) Lines 322-3 "muscarinic acetylcholine, cholinergic" do not make sense as descriptions of GPCR ligands; acetylcholine alone is a ligand

12) Line 331 dopamin acts a neurotransmitter in the brain, not a catecholamine

13) Lines 335-6 "Thus, few GPCRs agonists and antagonists have neuroprotective benefits and are considered drug targets". I think this should read "Thus, some GPCRs agonists and antagonists have neuroprotective benefits and their receptors are considered drug targets"

14) Line 369 "considered" not "ruminated"

 Answers: We have addressed Reviewer #1’s minor suggestions in the revised manuscript, which can be tracked.

Reviewer 2 Report

This well-written review comprehensively covers the topics from the basis of G-protein signaling to cardiovascular pathologies.  It is a good textbook also for non-specialist and students. However, the following points should be considered.

General comments:

(1) In the model introduced in this manuscript, ligand-free GPCR pre-couples with G-protein, and after ligand-binding, only G-alfa subunit is released. However, it is not the common mechanism of GPCR/G-protein interaction, because many GPCR binds to heterotrimeric G protein after ligand binding, and releases G-alfa and G-beta/G-gamma subunits after GDP/GTP exchange.

(2) Many readers would be interested in the pharmaceutical aspects of practical use. For example, Irbesartan (Avapro), which acts as an inhibitor of angiotensin-II receptor, is a popular drug for hypertension.

Minor points:

Line 102: "rhodopsin, secretin, glutamate, adhesion, and frizzled/taste2" should be "rhodopsin family, secretin family, glutamate family, adhesion family, and frizzled/taste2 family".

Lines 117, 143, 149, 150: "conformational" or "conformation"

Line 135: "Gas" ---> "G-alfa-s"

Line 136: "Gai" ---> "G-alfa-i"

Line 142: "legends" ---> "ligands"

Line 147: "ligands" ---> "agonists"

Line 152: ICL2 and ICL3 of GPCR are close to C-terminus, but distant from N-terminus.

Line 168: "GDP-GTP-exchange"?

Line 270: "functions" ---> "function"

Author Response

# Reviewer 2:

  • In the model introduced in this manuscript, ligand-free GPCR pre-couples with G-protein, and after ligand-binding, only G-alfa subunit is released. However, it is not the common mechanism of GPCR/G-protein interaction, because many GPCR binds to heterotrimeric G protein after ligand binding, and releases G-alfa and G-beta/G-gamma subunits after GDP/GTP exchange.

Answers: We have now modified Figures 1 & 2 to incorporate Reviewer #2’s suggestions. Kindly refer to Figures 1 & 2 of the revised manuscript.

  • Many readers would be interested in the pharmaceutical aspects of practical use. For example, Irbesartan (Avapro), which acts as an inhibitor of angiotensin-II receptor, is a popular drug for hypertension.

Answers: In response to Reviewer #2’s suggestions, we have included a new table in the revised manuscript (Please refer to Table 2).

Minor points:

Line 102: "rhodopsin, secretin, glutamate, adhesion, and frizzled/taste2" should be "rhodopsin family, secretin family, glutamate family, adhesion family, and frizzled/taste2 family".

Lines 117, 143, 149, 150: "conformational" or "conformation"

Line 135: "Gas" ---> "G-alfa-s"

Line 136: "Gai" ---> "G-alfa-i"

Line 142: "legends" ---> "ligands"

Line 147: "ligands" ---> "agonists"

Line 152: ICL2 and ICL3 of GPCR are close to C-terminus, but distant from N-terminus.

Line 168: "GDP-GTP-exchange"?

Line 270: "functions" ---> "function"

Answers: We have addressed Reviewer #1’s minor suggestions in the revised manuscript, which can be tracked.

Reviewer 3 Report

GPCRs are widely implicated in various cardiovascular biology and their regulation under different pathological conditions is highly complicated with an enormous amount of studies being done to date.  While the reviewer applauds the attempt to review such an important topic and did a reasonable job on the background of GPCR (sections 1-3). However, the intended scope of this manuscript to review GPCR in different cardiovascular diseases is over-ambitious. Specifically, it is very clear that sections 4.1-4.6, the core of this manuscript, are significantly underdeveloped and overall not very insightful. Much important literature seems to be missed in the review. Given there are many excellent existing review articles on GPCR in different cardiovascular diseases, the reviewer is unsure about the significance of this one in its current form.

Therefore, the reviewer thinks it's imperative to focus on just one pathophysiological condition and make sure all the important literature, classical and recent, is covered in the review with insightful interpretations. 

Author Response

# Reviewer 3:

  • Therefore, the reviewer thinks it's imperative to focus on just one pathophysiological condition and make sure all the important literature, classical and recent, is covered in the review with insightful interpretations. 

Answers: We agree with Reviewer #3 that the intended scope of this manuscript to review GPCR in different cardiovascular diseases is over-ambitious, but we have tried to justify it by incorporating new studies in the revised manuscript.  

Reviewer 4 Report

Cardiovascular diseases (CVDs) impose a huge health burden, and a large percentage of therapeutics target GPCRs. Hence it makes sense to review the regulation of G-proteins in cardiovascular diseases. While the subject matter is rich, this review in its current state lacks sufficient substance, making it unsuitable for publication.

A - Subject matter not discussed sufficiently / appropriately
(i) A large part of the review is dedicated to elaborating on GPCR signaling. While a small introduction / revision of GPCR signaling would be useful, presently almost 50% space is devoted to the same. 
(ii) In the sections where specific examples of CVDs are discussed, the text focusses on GPCRs at the expense of detailing the role of G-proteins in either the pathology or the symptoms of the disease. Examples detailed below. 

1.     Myocardial ischemia – no discussion of G proteins

2.     Hypertension – G proteins mentioned, but their contribution to pathology is not elaborated.

The cAMP dependent activation of PKC is a specific, limited pathway and more effort is needed to discuss the differences between cAMP-PKA and cAMP-PKC signaling axes.

3.     Atherosclerosis – no discussion of G proteins

4.     Stroke – an interesting example is provided of serotonin. As mentioned by the authors, serotonin receptors can couple to different flavours of G proteins. An explanation of what regulates or increases the deleterious signaling through Gi while reducing signaling through other G proteins is desirable. The same applies to the dopamine system.

5.     Peripheral arterial disease – no elaboration on the role of G proteins in pathology

6.     Restenosis – no elaboration on the role of G proteins in pathology

(iii) Some CVDs like PPHN – persistent pulmonary hypertension of the newborn, cardiomyopathy, cardiac hypertrophy, are interesting but have not been discussed.

(iv) In the beginning of the manuscript, large and small G proteins are listed. However, the discussion of pathology is limited to GPCRs and associated G proteins. Small-G proteins - Ras, Rac, Rho and Raf kinase- could also play role in specific CVDs or general dysfunction across CVDs. This needs to be discussed.

(v) Regulation of G proteins by post translational modifications (PTMs), and changes to PTMs in CVD could be considered.

The authors should either change the title to more closely reflect the matter discussed, or add matter that supports the title – viz. more details on G proteins and their role in CVD pathology.

B – Wrong terminology and Mischaracterization 

(i)     The following sentence (lines 43 -46) can be refuted “The interaction of ligands with their receptors and subsequent signaling processes are highly selective due to the specific structure of the receptors and their ligands (Key and lock mechanism).” There are many instances of multiplicity in GPCR signaling. First, one ligand is recognized by multiple receptors with different downstream effects based on the receptor type (eg. Alpha and beta adrenergic receptors can both bind to adrenaline). Second, the same receptor can recognize multiple ligands (eg. CCR5 can recognize CCL3, CCL4 and CCL5).

(ii)   The discussion of Galphai signaling (line 69 onwards) only mentions the inhibition of adenylyl cyclase by Galphai. Galphai and its associated beta-gamma subunits regulate various ion channels, including GIRKs (Jiang and Bajpayee, 2009, and references therein). 

(iii)  Simple, standard terms have been unnecessarily replaced – eg “hydrolyzation” in place of “hydrolysis” (line 91).

(iv)  The clause “Generally, GPCRs are odorant/pheromone receptors;” (line 97) seems incorrect – GPCRs detect a host of ligands including hormones, neurotransmitters, chemokines, compounds that impart taste (tastants) – just in mammals. They detect a variety of stimuli (bacterial folate, cAMP) in species such as Dictyostelium, and a different variety of compounds in fungi and plants.

(v)   In the figures, the ligand is shown bound to ECL3 of the GPCRs. It would be more appropriate to create a shaded region encompassing parts of all the ECLs, alongwith adjacent regions of the TM domains to indicate the diversity of ligand-binding pockets. 

C - Grammatical errors and improper sentence construction

(i)     The manuscript is replete with improper grammar or wrong choice of words. In the abstract itself, there are 3 instances. Underlined “Abstract: Cell signaling is a fundamental process that supports the cells to survive against various ecological and environmental signals and imparts tolerance against stressful conditions. The basic machinery for cell signaling includes a receptor molecule that senses and receives the signal. The primary form of signal might be hormone, light, antigen, odorant, neurotransmitter, etc. Similarly, heterotrimeric G-proteins principally impart communication from the plasma membrane via G-protein-coupled receptors (GPCRs) into the inner compartments of the cells to control various biochemical activities. These signals regulate different kinds of physiological functions by the targeted cell types. This review article discusses G proteins' signaling, regulation, and physiological activities. Also, we elaborated on the role of G proteins in several cardiovascular diseases, such as myocardial ischemia, hypertension, atherosclerosis, restenosis, stroke, peripheral artery disease, and restenosis.

(ii)   Mechanistically, contraction ligand proteins of Galphaq coupled GPCR,” (line 272)

(iii)  “Activated serotonin receptors stimulate Gi/Go-proteins that lead to harmful modulation of adenylyl cyclase, thereby reducing cAMP formation.” (lines 326 - 327).

(iv)  There is a difference in fonts across references. 

I strongly recommend that the authors use a professional editing service to improve the readability of the manuscript. 

Summary 

In its present form the manuscript contains errors and omissions. It is not comprehensive wrt GPCR signaling, neither is it detailed in its discussion about the involvement of G-proteins in CVDs. Publishing this review in its present form, would not be beneficial to the scientific community, nor would it serve the purpose of the journal.

Author Response

# Reviewer 4:

A - Subject matter not discussed sufficiently / appropriately

  • A large part of the review is dedicated to elaborating on GPCR signaling. While a small introduction / revision of GPCR signaling would be useful, presently almost 50% space is devoted to the same. 

Answer: In response to Reviewer #4’s suggestions, we have modified the title and included GPCRs.

  • In the sections where specific examples of CVDs are discussed, the text focusses on GPCRs at the expense of detailing the role of G-proteins in either the pathology or the symptoms of the disease. Examples detailed below. 

Answers: In response to Reviewer #4’s suggestions, we have now incorporated them in the revised manuscript (please refer to Page 10, lines 1516-1547, Page 11, lines 1880-1888; Page 12, lines 2171-2183; Page 13 &14, lines 2389-2619).

  • Some CVDs like PPHN – persistent pulmonary hypertension of the newborn, cardiomyopathy, cardiac hypertrophy, are interesting but have not been discussed.

Answers: We agree with Reviewer #4 that we have not discussed the listed CVDs. In the present review, we have discussed the role of G-proteins and GPCRs in some important cardiovascular diseases, and we also agree with reviewer #3 that discussing the role of G-proteins and GPCRs in every cardiovascular disease is over-ambitious.

  • In the beginning of the manuscript, large and small G proteins are listed. However, the discussion of pathology is limited to GPCRs and associated G proteins. Small-G proteins - Ras, Rac, Rho and Raf kinase- could also play role in specific CVDs or general dysfunction across CVDs. This needs to be discussed.

Answers: We agree with Reviewer #4 that we have discussed and listed small G-proteins in the manuscript, but it is to provide a comprehensive picture to the reader about various G-proteins. Till now, more than 100 small G proteins have been identified and are structurally divided into subfamilies: Ras (Ras, Rap, Rad, Ral, Rin, and Rit), Rho (Rho, Rac, Cdc42, and Rnd), Rab, Sar1/ADP ribosylation factor (Arf, Arl, Ard, and Sarl), and Ran. The subfamilies' function in the regulation of gene expression, cell proliferation and migration, vesicle trafficking, and transportation are discussed in numerous manuscripts, and it is beyond the scope of this manuscript to include the role of these small G-proteins in cardiovascular diseases. 

  • Regulation of G proteins by post translational modifications (PTMs), and changes to PTMs in CVD could be considered.

Answers: In response to Reviewer #4’s suggestions, we have incorporated it in the revised manuscript (please refer to section 3.1.1., Pages 3 & 4, lines 302-441).

B – Wrong terminology and Mischaracterization 

(i)     The following sentence (lines 43 -46) can be refuted “The interaction of ligands with their receptors and subsequent signaling processes are highly selective due to the specific structure of the receptors and their ligands (Key and lock mechanism).” There are many instances of multiplicity in GPCR signaling. First, one ligand is recognized by multiple receptors with different downstream effects based on the receptor type (eg. Alpha- and beta-adrenergic receptors can both bind to adrenaline). Second, the same receptor can recognize multiple ligands (eg. CCR5 can recognize CCL3, CCL4 and CCL5).

(ii)   The discussion of Galphai signaling (line 69 onwards) only mentions the inhibition of adenylyl cyclase by Galphai. Galphai and its associated beta-gamma subunits regulate various ion channels, including GIRKs (Jiang and Bajpayee, 2009, and references therein). 

(iii)  Simple, standard terms have been unnecessarily replaced – eg “hydrolyzation” in place of “hydrolysis” (line 91).

(iv)  The clause “Generally, GPCRs are odorant/pheromone receptors;” (line 97) seems incorrect – GPCRs detect a host of ligands including hormones, neurotransmitters, chemokines, compounds that impart taste (tastants) – just in mammals. They detect a variety of stimuli (bacterial folate, cAMP) in species such as Dictyostelium, and a different variety of compounds in fungi and plants.

(v)   In the figures, the ligand is shown bound to ECL3 of the GPCRs. It would be more appropriate to create a shaded region encompassing parts of all the ECLs, alongwith adjacent regions of the TM domains to indicate the diversity of ligand-binding pockets. 

Answers: We have addressed all of Reviewer #4’s suggestions in the revised manuscript, and they can be tracked.

C - Grammatical errors and improper sentence construction

(i)     The manuscript is replete with improper grammar or wrong choice of words. In the abstract itself, there are 3 instances. Underlined “Abstract: Cell signaling is a fundamental process that supports the cells to survive against various ecological and environmental signals and imparts tolerance against stressful conditions. The basic machinery for cell signaling includes a receptor molecule that senses and receives the signal. The primary form of signal might be hormone, light, antigen, odorant, neurotransmitter, etc. Similarly, heterotrimeric G-proteins principally impart communication from the plasma membrane via G-protein-coupled receptors (GPCRs) into the inner compartments of the cells to control various biochemical activities. These signals regulate different kinds of physiological functions by the targeted cell types. This review article discusses G proteins' signaling, regulation, and physiological activities. Also, we elaborated on the role of G proteins in several cardiovascular diseases, such as myocardial ischemia, hypertension, atherosclerosis, restenosis, stroke, peripheral artery disease, and restenosis.”

(ii)   “Mechanistically, contraction ligand proteins of Galphaq coupled GPCR,” (line 272)

(iii)  “Activated serotonin receptors stimulate Gi/Go-proteins that lead to harmful modulation of adenylyl cyclase, thereby reducing cAMP formation.” (lines 326 - 327).

(iv)  There is a difference in fonts across references. 

I strongly recommend that the authors use a professional editing service to improve the manuscript's readability. 

 Answers: We have used MDPI professional editing services, and the certificate is attached herewith.

Round 2

Reviewer 1 Report

The authors have acted on my suggestions and I am satisfied that the review is substantially improved. There are a few small issues that need to be dealt with.

Line 45; second messengers, not secondary messengers

Line 134, 3.1.1. G-proteins post- translational modification. This section is entirely about modification of GPCRs, not G proteins; the subheading needs to be changed. 

Lines 174-180. The authors now talk more about biased agonists, but I feel these need more introduction. I would recommend addings a sentence to define a biased agonist at this point, where the different classes of ligands for GPCRs are first mentioned.

Line 553 Table 2. "Disease" or "Condition" would be a better heading for column 2 rather than "manifestation"

Line 661 Gbg; this should be Gbg

Author Response

We are thankful for your kind decision letter and the Reviewers’ comments on our manuscript entitled “Role of G-proteins and GPCRs in Cardiovascular Pathologies.” Based on Reviewers’ comments or suggestions, we have revised the manuscript and incorporated the changes in the manuscript as tracked changes.

# Reviewer 1:

  • Line 45; second messengers, not secondary messengers

Answers: In response to Reviewer #1’s suggestions, we have corrected it in the revised manuscript.

  • Line 134, 3.1.1. G-proteins post- translational modification. This section is entirely about modification of GPCRs, not G proteins; the subheading needs to be changed..

Answers: In response to Reviewer #1’s suggestions, we have now incorporated G-proteins post-translational modification in the revised manuscript (please refer to Page 3, lines 141-149; Page 4, lines 150-169).

  • Section 3.1.2, " Lines 174-180. The authors now talk more about biased agonists, but I feel these need more introduction. I would recommend adding a sentence to define a biased agonist at this point, where the different classes of ligands for GPCRs are first mentioned.

Answers: In response to Reviewer #1’s suggestions, we have now incorporated it in the revised manuscript (please refer to Page 3, lines 101-107).

  • Table 2. "Disease" or "Condition" would be a better heading for column 2 rather than "manifestation"

Answers: We have corrected it in the revised manuscript.

  • Line 661 Gbg; this should be Gbg

Answers: We have now corrected it in the revised manuscript.

We thank the reviewer for their suggestions; the manuscript has improved considerably.

Reviewer 3 Report

The authors have addressed all my comments and I have no additional concerns.

Author Response

We addressed all the the reviewer #3 concerns and the manuscript has improved considerably. 

Reviewer 4 Report

The present version is definitely better than the original submission. However, there seem to be some omissions that need to be addressed. Additionally, better proof-reading is definitely required. In summary, some/ not-minor rewriting is recommended before acceptance.

1.     One of the major issues with the previous version was the lack of discussion of the role of G proteins in CVD pathology. This major point has been adequately addressed. 

2.     I would direct the authors to re-examine lines 135 onwards, page 3. There seems to have been an intention to include discussion on the post-translational modification of G-proteins, and sub-sections have been created. However, the titles of these sub-sections have not been populated.  

3.     Organization – Some parts seem out of place in the reading of the review. Eg. Line 172 “This receptor superfamily displays 7TM helical domains linked by the alternation of three intracellular and three extracellular loops, such as ICL1, ECL1, ICl2, ECL2, ICL3 and ECL3).” – is out of place in the section “3.1.4- GPCR regulators”. Lines 287-291 on page 9 are out of context. Similarly, in the discussion of myocardial ischemia, the authors start discussing beta-adrenergic receptors and adenylate cyclase, then take a detour to AT1R (line 395-400, page 13) and then come back to adenylate cyclase.

4.     Minor issues – words have been repeated – “restenosis” appears twice in the abstract.

Line 661, the symbols “bg” are seen in place of the Greek symbols for beta and gamma. 

Author Response

We are thankful for your kind decision letter and the Reviewers’ comments on our manuscript entitled “Role of G-proteins and GPCRs in Cardiovascular Pathologies.” Based on Reviewers’ comments or suggestions, we have revised the manuscript and incorporated the changes in the manuscript as tracked changes.

# Reviewer 4:

  • I would direct the authors to re-examine lines 135 onwards, page 3. There seems to have been an intention to include discussion on the post-translational modification of G-proteins, and sub-sections have been created. However, the titles of these sub-sections have not been populated. 

Answer: In response to Reviewer #4’s suggestions, we have now incorporated G-proteins post-translational modification in the revised manuscript (please refer to Page 3, lines 141-149; Page 4, lines 150-169).

  • Organization – Some parts seem out of place in the reading of the review. Eg. Line 172 “This receptor superfamily displays 7TM helical domains linked by the alternation of three intracellular and three extracellular loops, such as ICL1, ECL1, ICl2, ECL2, ICL3 and ECL3).” – is out of place in the section “3.1.4- GPCR regulators”. Lines 287-291 on page 9 are out of context. Similarly, in the discussion of myocardial ischemia, the authors start discussing beta-adrenergic receptors and adenylate cyclase, then take a detour to AT1R (line 395-400, page 13) and then come back to adenylate cyclase.

Answers: In response to Reviewer #4’s suggestions, we have now corrected it in the revised manuscript (please refer to Page 3, lines 110-112; Page 10, lines 344-346; Page 10, lines 367-372 ).

  • Minor issues – words have been repeated – “restenosis” appears twice in the abstract.

Line 661, the symbols “bg” are seen in place of the Greek symbols for beta and gamma

Answers: We have now corrected it in the revised manuscript.

We thank the reviewer for their suggestions; the manuscript has improved considerably.